# “It’s All COVID’s Fault!”: Symptoms of Distress among Workers in an Italian General Hospital during the Pandemic

**DOI:** 10.3390/ijerph19127313

**Published:** 2022-06-14

**Authors:** Michele Mastroberardino, Riccardo Cuoghi Costantini, Antonella Maria Pia De Novellis, Silvia Ferrari, Costanza Filippini, Fedora Longo, Mattia Marchi, Giulia Rioli, Laura Valeo, Roberto Vicini, Gian Maria Galeazzi, Roberto D’Amico, Paola Vandelli

**Affiliations:** 1Servizio Formazione, Ricerca e Innovazione, Azienda Ospedaliero-Universitaria di Modena, Via del Pozzo 71, 41124 Modena, Italy; riccardo.cuoghicostantini@unimore.it (R.C.C.); roberto.vicini@unimore.it (R.V.); roberto.damico@unimore.it (R.D.); vandelli.paola@aou.mo.it (P.V.); 2Department of Medical and Surgical Sciences for Mother, Child and Adult, University of Modena and Reggio Emilia, Via del Pozzo 71, 41124 Modena, Italy; 3Department of Biomedical, Metabolic and Neural Sciences, School of Specialization in Psychiatry, University of Modena & Reggio Emilia, Via G. Campi 287, 41125 Modena, Italy; amp.denovellis@gmail.com (A.M.P.D.N.); silvia.ferrari@unimore.it (S.F.); 187161@studenti.unimore.it (C.F.); lally.v@hotmail.it (L.V.); gianmaria.galeazzi@unimore.it (G.M.G.); 4Dipartimento ad attività integrata di Salute Mentale e Dipendenze Patologiche, Azienda USL-IRCCS di Reggio Emilia, Via G. Amendola 2, 42122 Reggio Emilia, Italy; mattia.marchi@unimore.it (M.M.); giulia.rioli@ausl.re.it (G.R.); 5Dipartimento di Salute Mentale e Dipendenze Patologiche, USL di Modena, Via L.A. Muratori 201, 41124 Modena, Italy; f.longo@ausl.mo.it; 6Department of Biomedical, Metabolic and Neural Sciences, PhD School in Neurosciences, University of Modena & Reggio Emilia, Via G. Campi 287, 41125 Modena, Italy; 7Department of Biomedical, Metabolic and Neural Sciences, International PhD School in Clinical and Experimental Medicine, University of Modena & Reggio Emilia, Via G. Campi 287, 41124 Modena, Italy

**Keywords:** COVID-19, pandemic, healthcare workers, distress, depression, anxiety, resilience

## Abstract

Background: Since the outbreak of the COVID-19 pandemic, healthcare workers (HCWs) have been faced with specific stressors endangering their physical and mental health and their functioning. This study aimed to assess the short-term psychological health of a sample of Italian HCWs and the related influencing factors. In particular, the study focused on the differences related to HCWs’ gender and to having been directly in charge of COVID-19 patients or not. Methods: An online survey was administered to the whole staff of the Modena General University Hospital three months after the onset of the pandemic, in 2020. Demographic data and changes in working and living conditions related to COVID-19 were collected; mental health status was assessed by the Depression, Anxiety and Stress Scale (DASS-21) and the Impact of Event Scale-Revised (IES-R). Results: 1172 out of 4788 members returned the survey (response rate = 24.5%), the male/female ratio was 30/70%. Clinically significant symptoms assessed according to the DASS-21 emerged among 21.0% of the respondents for depression, 22.5% for anxiety and 27.0% for stress. Symptoms suggestive of a traumatic reaction were reported by 19.0% of the sample. Symptoms of psychological distress were statistically associated with female gender, job role, ward, changes in lifestyle, whereas first-line work with COVID-19 patients was statistically associated with more stress symptoms. HCWs reported a significant level of psychological distress that could reach severe clinical significance and impact dramatically their quality of life and functioning. Conclusions: Considering the persistence of the international emergency, effective strategies to anticipate, recognize and address distress in HCWs are essential, also because they may impact the organization and effectiveness of healthcare systems.

## 1. Introduction

Severe Acute Respiratory Syndrome Corona Virus 2 (SARS-CoV-2) is the infective agent responsible for COVID-19 (Corona Virus Disease 19). COVID-19 primarily affects the low respiratory tract and causes a variety of flu-like symptoms, including fever, cough, shortness of breath, muscular pain, fatigue, changes in gustatory and olfactory sensitivity (anosmia and dysgeusia) and gastrointestinal disorders such as diarrhea [1]. In severe forms, pneumonia may occur, with acute respiratory distress syndrome, sepsis and septic shock, even death. Cardio-circulatory, thromboembolic, neurological and inflammatory complications may also occur. Advanced age and co-morbidities are identified as predictive factors of evolution towards severe forms of the disease [2]. In March 2020, the World Health Organization (WHO) defined COVID-19 as a pandemic [3], and countries around the world began enacting measures with the aim to contain the emergency. These included, but were not limited to, social distancing (the so-called “quarantine” and “lockdown”), implementation of personal protective equipment (PPE), increase of resources to support public health services [4]. At the time of writing, several vaccines have been developed, and vaccine campaigns have started throughout the world, with positive results [5,6]. New variants of SARS-CoV-2, though, are rapidly spreading, causing the persistence of uncertainties and risks [7,8]. On 1 May 2021, the WHO declared more than 153 million confirmed cases of COVID-19 and 3.2 million of COVID-related deaths [3]. The COVID-19 pandemic has also conveyed a large psycho-social backlash [9,10], particularly impacting the most vulnerable fringes of society, such as the elders, children, teenagers, and people with pre-existing physical or psychological diseases [11,12]. A recent study conducted in Italy [13] investigated how the COVID-19 pandemic and lockdown measures impacted on perceived quality of life (QoL) in a large sample from the general population: women reported overall worse psychological, physical, and environmental QoL compared to men, and the lowest scores were found in the dimensions of life satisfaction and pleasure. These results were confirmed in another recent Italian work [14]. Of interest, young adults emerged as the most psychologically vulnerable subjects. The same was shown in a recent meta-analysis conducted in Canada [15]. Among the risk factors predicting a poor perceived QoL, there were lower education levels, being unemployed or university students, suffering from pre-existing psychiatric and medical syndromes, having a job activity suspended, and not completely adhering to the recommended measures against infection with SARS-CoV-2. High levels of PTSD symptoms were found (i.e., up to 29.5%) in an Italian population sample, suggesting that the COVID-19 pandemic should be considered as a proper traumatic event [16]. Since the earliest beginning of the pandemic, research aimed at identifying viable health policies to provide specific psychological support to the affected populations. The exponential increase in healthcare needs, mostly required for urgent conditions, within both hospitals and communities, resulted in work overload for healthcare workers (HCWs). That warranted concerns about health consequences in this working category, especially in those operating at the front line, also for the risk of contracting the infection [17,18]. The concern was further increased by reports on the number of COVID-19 victims among HCWs, mostly medical doctors and nurses [19]. An increased risk to develop stress-related and affective conditions was also acknowledged: HCWs reported symptoms related to depression, anxiety, stress, insomnia, somatization, obsessive-compulsive disorder, PTSD, with front-line HCWs, nurses and females reporting even higher levels of distress [20,21,22,23,24,25]. One of the first analyses was conducted in Wuhan, the first site of the SARS-CoV-2 outbreak, and showed a 72% positivity of HCWs at the IES-R scale, documenting stress-related and post-traumatic effects [26]. Contrasting results came from a study in Singapore, showing reduced levels of depression, anxiety, distress and PTSD among HCWs in comparison to those during past epidemics, and higher distress among non-medical staff [27]. An Italian study on 1379 HCWs found high levels of symptoms related to PTSD (49.38%), depression (24.73%), anxiety (19.80%), insomnia (8.27%) and stress (21.90%) [28]. Many factors related to the exceptional working conditions imposed by the pandemic were addressed as relevant in causing emotional disruption in HCWs, especially those at the front line: stigmatization, fatigue due to strict bio-security measures, major demands in the workplace (long working hours, increased number of patients and maintenance of best practices, need to replace a sick colleague), reduced ability to use social support due to working time, insufficient or controversial information, concern about the risk of possible transmission of the SARS CoV-2 infection to one’s family, leading to stigma or self-stigma [29]. Similar reports from past infectious epidemics, such as SARS and Ebola, despite the lesser epidemiological entity, confirmed that the rates of PTSD increased among HCWs at the forefront of the emergency [30,31]. Awareness of the need for adequate psychological monitoring and support to HCWs has gradually developed as the pandemic has evolved, up to the present days. Conti et al. [32] found that 39.3% of subjects in a sample of 933 HCWs expressed the need for psychological support. Similarly, 51.0% of highly stressed HCWs felt the need for psychological support, and 65.9% of these individuals were part of the medical staff [24]. Evidence of the potential benefits of psychological interventions, even basic ones, is increasingly available: the IES-R scores of a sample of high-risk HCWs statistically significantly decreased after 2 weeks of telephone-based psychiatric consultations [33]. Another study conducted in Shanghai described the appropriate psycho-social interventions, which should include strategic planning and coordination of psychological first aid, a crisis prevention and response system, epidemiological reporting and targeted interventions [34]. The WHO also divulged a set of recommendations, underlying that the mental health and psycho-social well-being of HCWs are as important as their physical health. Among these, there were: adequate rest and breaks during work or between shifts, high-quality nutrition, physical activity, staying in touch with family and friends, avoiding the use of tobacco and alcohol, appropriate use of social media and telephone, appropriate sharing of common experiences and asking for psychological support. Emilia-Romagna, in the North of Italy, was one of the areas over the Italian territory first and most affected by the pandemic and its consequences on the healthcare system [35], and one of the most affected in Europe. Recently, different studies have investigated the short- and medium-term impact of the COVID-19 pandemic on the mental health of HCWs working in Italian healthcare facilities. One study included the HCWs of three hospitals in Milan, Italy [36], another aimed at assessing the mental health status of Italian rehabilitation HCWs in a Southern Italy University Hospital [37], another is a multicentric national study including five major University hospitals from three different regions affected at different degrees of severity by the outbreak, i.e., Tuscany, with a low level of exposure, Emilia-Romagna (Bologna and Ferrara), with a medium level of exposure, and Lombardy, with a high level of exposure [38].

Both through our review of the scientific evidence and as a consequence of direct clinical work in the field of prevention and welfare of HCWs, we identified some knowledge gaps that enabled us to define the following purposes of the study: Primary objective: to assess the prevalence and incidence of symptoms of mental health distress in a sample of HCWs working in two general hospitals in Modena, Emilia Romagna, Italy;Secondary objective: to assess the risk and protective factors affecting the mental health of the respondents, with special attention to the presumed role of gender and of having been directly responsible for COVID-19 patients.

## 2. Materials and Methods

### 2.1. Study Design

#### Observational, Cross-Sectional Study

##### Study Procedure, Setting, Data Collection 

A survey was developed and administered to the entire working population of the two main public hospitals in the city of Modena. The two University-based General hospitals share a common administration and organization (Azienda Ospedaliero-Universitaria, AOU); with their cumulative 1108 beds and more than 45,000 admissions per year, they provide an occupation for more than 4000 workers. 

Inclusion criteria were: working at the AOU of Modena;having understood and accepted the terms of the informed consent;being able to read and understand the Italian language.

An online, electronic survey was developed by means of the LimeSurvey software (free access for research purposes) and distributed via email to all subjects working in the two hospitals, between May and August 2020. In the email, the text of the message provided information about the study and an invitation to participate, including the link to the webpage containing the survey. By clicking on the link, the respondents also expressed their consent to participate. Several reminders were sent via email to the non-respondents, every 2 weeks until the month of August 2020. Each professional could participate in the survey only once.

### 2.2. Measures

The questionnaire developed for the survey included:

A personal data sheet, collecting information on sex, age, professional role, type of contract, department/service, educational level, cohabitation and children, working seniority; Specific questions related to the experience during the emergency, such as adjustments required to the living or working situation, drinking, smoking and eating habits, personal or family experience with the infection, experience of death caused by COVID-19;The following psychometric tools:1.The Depression Anxiety Stress Scale—Short Version (DASS-21), in its Italian validated version [39,40]. This is a self-rating, 21-item scale, with a 4-point Likert scale (0 = Does not apply to me at all to 3 = Applies a lot or most of the time to myself), investigating the 3 symptomatic dimensions of Depression, Anxiety and Stress. The Depression subscale assesses symptoms such as dysphoria, hopelessness, self-worthlessness and lack of interest; the Anxiety subscale comprises items evaluating somatic and psychic symptoms of anxiety and situational anxiety; the Stress subscale appraises a condition of persistent arousal and tension, with relaxation difficulties, impatience, irritability and restlessness [41]. The DASS-21 provides a final total score ranging from 5 to ≥14 in the Depression dimension, from 4 to ≥10 in the Anxiety dimension, from 8 to ≥19 in the Stress dimension. The score may suggest mild to extremely severe symptoms in each dimension, according to standardized cut-off scores. Scores were aggregated into binary categories, from moderate to extremely severe (presence of moderate to extremely severe symptomatology) including subjects scoring ≥14 for the Anxiety dimension; ≥10 for the Depression dimension; ≥19 for the Stress dimension;2.The Impact of Event Scale-Revised (IES-R), in its Italian validated version [42,43]. This is a 22-item, self-rating standardized psychometric scale, used to investigate the presence of post-traumatic symptoms. This tool is composed of three sub-dimensions (Re-experience, Hyperarousal, Avoidance). The respondents must rate each item on a scale from 0 (Not at all) to 4 (Extremely), based on their experience with respect to the traumatic event, referring to the last 7 days. The IES-R is a useful assessment to quantify the reactions to stress after a series of traumatic events and has been found to be a valuable tool for identifying individuals who would require specialist intervention (Horowitz et al., 1979). The scores were aggregated into binary categories, with moderate–severe including scores ≥33. 

### 2.3. Statistical Analysis 

Descriptive and inferential analyses were performed according to the nature of the variables (continuous or categorical). No power calculation to pre-define the required dimension of the sample was considered necessary, since the study aimed at reaching virtually the whole working population of the two hospitals. Scores from the psychometric tools (DASS-21 and IES-R) were dichotomized and used as dependent variables in the logistic regression model, aiming at recognizing possible protective and risk factors. The association was expressed in terms of odds ratio (OR) with a 95% confidence interval (95%CI). Results with p-values less than 0.05 (5%) were considered statistically significant. Statistical analyses were performed by using the Stata^®^ 15 software.

## 3. Results

Considering, at the time of conduction of the study, a total working population in the hospitals of 4788 subjects, valid responses were collected from 1172 HCWs (response rate of 24.5%); of these, 70.3% were female (n = 824), with most subjects with an age ranging from 45 to 54 (n = 348, 29.7%). Table 1 includes all the data collected with the survey. Figure 1 specifically reports changes in lifestyle and behavior. 

The table also includes, in the bottom lines, the number of subjects achieving clinical significance (score from moderate to severe/extremely severe) at the two psychometric measures and their proportion considering the total number of respondents, which is also detailed in Figure 2. At the psychometric assessment, the following average scores were determined (± standard deviations): DASS-21 depression = 7.80 ± 8.14; DASS-21 anxiety = 5.54 ± 6.69; DASS-21 stress = 13.78 ± 9.18; cumulative IES-R = 19.08 ± 15.89. 

HCWs working in the COVID-19 areas were 14.8% (n = 173) of the respondents and worked in the following wards/units: intensive care; emergency room and trauma-emergency coordination; emergency medicine; infectious diseases; respiratory medicine. 

Table 2 includes statistically significant associations between the results at the psychometric assessment and other clinical and non-clinical variables.

Specifically, high levels of stress, anxiety and depression according to the DASS-21 were significantly related to female gender, increased consumption of alcohol and cigarettes and a less healthy diet. In addition, high levels of stress and anxiety were found to be related to increased workload and to the professional role of manager.

High levels of DASS-21-anxiety were also significantly associated with high-school educational level, while a higher DASS-21-stress score was found among those workers directly caring for COVID-19 patients, among workers at the diagnostic imaging department and among members of the general direction of the hospital.

Finally, female gender, increased consumption of alcohol and cigarettes, an unregulated diet, increased workload, a university-level education and the professional role were significantly associated with higher scores at the IES-R.

## 4. Discussion

The present study aimed at assessing the short-term prevalence of mental health problems in a sample of Italian HCWs after the onset of the COVID-19 pandemic and to recognize the involved risk and protective factors. In particular, the study focused on the differences related to gender and to being or not in direct contact with COVID-19 patients. 

Briefly, a significant correlation between female sex and high levels of depression, anxiety and stress was confirmed by the results at the three subscales of the DASS-21 and at the IES-R. Among HCWs working in COVID-19 areas, symptoms of anxiety and stress were also significantly higher, while no significant correlations were found with symptoms of depression or PTSD.

In our sample, three months after the outbreak of the COVID-19 pandemic, about one HCW out of five reported clinically significant symptoms of depression, anxiety, and PTSD, whereas one out four displayed above-threshold symptoms of stress in the DASS-21. 

The study by Magnavita et al. [44] suggested that HCWs had a four-fold higher risk of developing depressive and anxious symptoms if they were working with COVID-19 patients and if they had contracted COVID-19. A more recent retrospective study involving physicians, physical therapists and nurses working in rehabilitation in a Southern Italy University Hospital showed significant mental health worsening among HCWs during the COVID-19 pandemic, with female HCWs reporting a higher risk of mental health problems [37]. Our findings complement this evidence.

With respect to PTSD rates among our sample, it could be argued that working in the hospital at the time of the pandemic exposed HCWs to a traumatic experience, including the exposure to the suffering and death of many patients, while perceiving the same risk for themselves and their families at home. These findings are coherent with similar literature data regarding previous epidemics, such as SARS, Ebola and H1N1 [30,45,46,47,48]. Although it may be supposed that HCWs working closely with COVID-19 patients were exposed to more traumatic events [49,50], in our sample the risk of PTSD was not higher among these HCWs, who however reported significantly higher rates of anxiety and stress compared to their colleagues not working with COVID-19 patients. This may be explained by considering that the survey was administered rather soon after the outbreak of the COVID-19 pandemic, thus measuring its psychological impact in the short term: anxiety and stress are more likely to be related to an immediate response to a stimulus, while depressive and post-traumatic symptoms could be related to a prolonged exposure or develop in the long run. Our findings are similar to evidence from Oman which showed that front-line HCWs were 1.5 times more likely to report anxiety, stress and insomnia, but not depression, as compared with colleagues not working in COVID-19 units [20].

In line with previous research [51], we found that the incidence of PTSD, measured by the IES-R scale, was statistically significantly higher among women. An Italian study confirmed these findings among Italian nurses [52]. The level of severe to moderate symptoms of depression and anxiety was also found to be higher among females HCWs, which confirms a body of previous research [26,32,50,53,54,55].

Some behavioral disturbances or non-adaptive behaviors were also described in our sample: a statistically significant correlation was found between the increase in alcohol and cigarette consumption and the presence of symptoms suggestive of depression, anxiety, stress and PTSD; this correlation also emerged with the adoption of more unregulated eating habits. Notably, food (especially “junk” or, anyway, unhealthy or excessive in quantity), alcoholic beverages and cigarettes may be part of a strategy of self-care, to alleviate the psychological distress associated, for example, with a complex work or life situation. Our results are similar to those of a 2020 Belgian study on the general population that highlighted a self-reported increase in alcohol consumption and cigarettes smoking during the COVID-19 lockdown, but opposite to those of a survey among Turkish HCWs, that showed high levels of depression, anxiety and stress levels according to the DASS-21, but also decreased daily consumption of cigarettes and alcohol [56,57]. 

A recent systematic review including 86 studies on the psychological and/or psychosomatic impact in HCWs caring for patients affected by different infectious diseases (SARS, H1N1, Ebola, Middle East respiratory syndrome and COVID-19) found an increased risk for stress, sleeping difficulties, depressive symptoms, anxiety, PTSD and somatization among front-line professionals. These findings show the high risk of working at the first line in emergency conditions. Undoubtedly, considering its world-wide diffusion and duration, the COVID-19 pandemic has had a stronger impact on HCWs, compared with previous pandemics [58]. 

### Limitations

A few limitations of the present study have to be acknowledged. First, at present, the data only provide a static description of the sample, at one given moment, therefore making causative assumptions impossible. Nevertheless, they may still provide useful information that will be further integrated as soon as data from the follow-up will be available. The response rate of the survey was rather low, impacting on the final dimension of the sample, and it may be that distress is in itself the reason why some of the HCWs refused to be involved. Still, this was the best available way to have a quick and feasible picture of the situation, that may help structuring interventions and other initiatives dedicated to HCWs. 

Another limitation derives from the choice of non-random data for the statistical analysis, since respondents with biases may include themselves into the sample, inducing a selection bias. Hence, these findings cannot be generalized and may therefore mislead.

Finally, we acknowledge the exploratory nature of the present study; more specifically, we are aware that the choice of univariate tests on individual response variables, despite having the advantage of simplicity of interpretation, fails to account for the covariance/correlation in the data. Multivariate statistical techniques including analysis of covariance and correction for confounders are needed to more adequately capture the multidimensional pattern of psychological distress in HCWs.

## 5. Conclusions

Our results confirm that HCWs have experienced high levels of emotional sufferings as a consequence of the COVID-19 pandemic, with female gender and being at the front-line of care of COVID-19 patients as considerable risk factors for clinically significant symptoms. So far, health care systems across the world have been massively stressed in the different phases of the pandemic and may still be under pressure in the future. Not only significant ameliorations are needed in terms of organization and distribution of resources, but also effective interventions to support the health of HCWs, with special attention to psychic distress, anxiety, depression, and PTSD. Further investigations are required to analyze the long-term effects of the current pandemic on the mental health of HCWs and to elucidate vulnerability factors associated with the development of distress. A more proactive attitude, aimed at increasing resilience and coping strategies, is also welcome and should be supported by scientific evidence.

## Figures and Tables

**Figure 1 ijerph-19-07313-f001:**
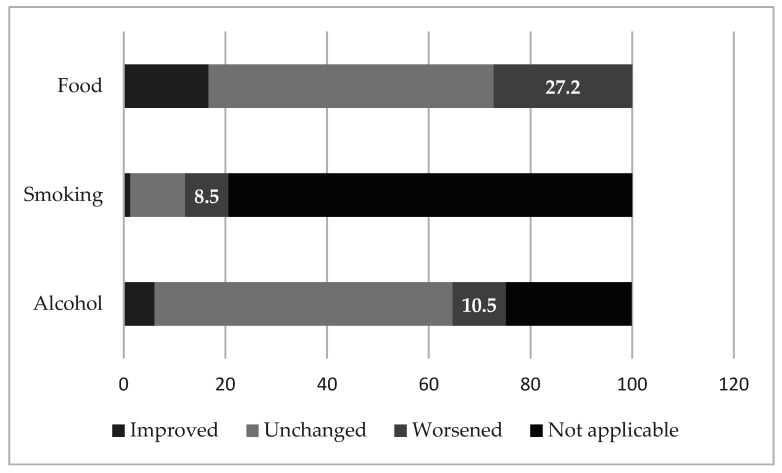
Changes in lifestyle. “Not applicable” refers to respondents who reported not to drink alcohol or to smoke.

**Figure 2 ijerph-19-07313-f002:**
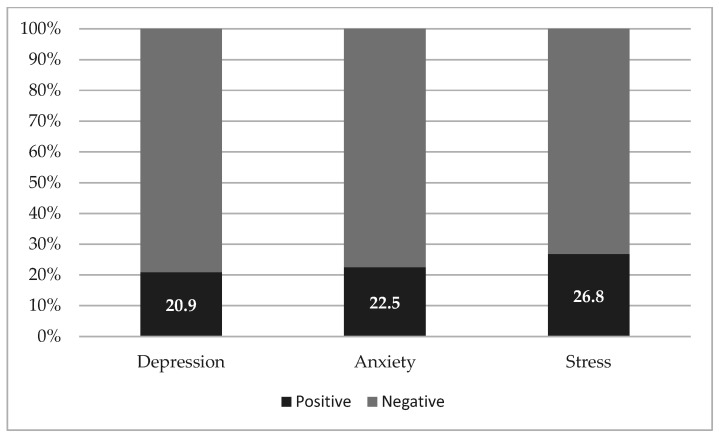
Psychometric assessment.

**Table 1 ijerph-19-07313-t001:** General summary of the results.

Variable	N = 1172	%
**(a)** **Socio-demographic**		
Gender		
M	348	29.7
F	824	70.3
Age range (years)		
≤34	298	25.4
35–44	229	19.5
45–54	348	29.7
≥55	297	25.3
Educational level (years of schooling)		
Elementary/Intermediate	41	3.5
High-school Diploma	239	20.4
Bachelor’s Degree	96	8.2
Master Degree or higher	796	67.9
Living situation		
With partner and children	517	44.1
With partner	280	23.9
Alone	193	16.5
With parents/siblings	87	7.4
Alone with children	63	5.4
Co-housing	27	2.3
Other	5	0.4
**(b)** **Working situation**		
Professional role		
Other managers	513	43.8
Nurses, sanitary technicians	464	39.6
Administration	128	10.9
Physicians and surgeons	67	
Type of contract		
Open-ended	896	76.4
Residents	184	15.7
Fixed-termed	68	5.8
Freelance	24	2.1
Department/Service		
Direction and staff	275	23.5
General surgery and surgical specialties	143	12.2
Neuroscience and head-neck	116	9.9
Specialized internal medicine	114	9.7
General internal medicine	106	9.0
Oncology and haematology	86	7.3
Obstetrics and gynaecology	75	6.4
Emergency	68	5.8
Nephrology and cardiology	62	5.3
Diagnostic imaging	51	4.4
Labs/pathological anatomy	42	3.6
Orthopaedics	31	2.6
Pharmacy	3	0.3
Seniority in employment (years)		
≤5	420	35.8
6–20	331	28.2
21–30	232	19.8
≥31	189	16.1
**(c)** **Working situation during the COVID-19 pandemics**		
Having direct contact with COVID-19 patients		
Yes	565	51.8
No	607	48.2
Being in charge of COVID-19 patients		
Direct (“front-line HCWs”)	173	14,8
Occasional	999	85,2
Increase in workload		
Yes	654	55.8
No	518	44.2
Infected with COVID-19		
Yes	54	4.6
No	1118	95.4
Infection among close relatives		
Yes	81	6.9
No	1091	93.1
Changes in the use of alcoholics		
No variation	687	58.6
Yes, decreased	72	6.1
Yes, increased	123	10.5
I don’t drink	290	24.7
Changes in smoking		
No variation	127	10.8
Yes, decreased	15	1.3
Yes, increased	100	8.5
I don’t smoke	930	79.4
Changes in eating habits		
No variation	657	56.1
Yes, improved	196	16.7
Yes, worsened	319	27.2
**(d)** **Psychometric assessment**		
DASS-21 depression ^§^		
Positive	245	20.9
Negative	927	79.1
DASS-21 anxiety ^§§^		
Positive	264	22.5
Negative	908	77.5
DASS-21 stress ^§§§^		
Positive	314	26.8
Negative	858	73.2
IES-R ^§§§§^		
Positive	219	18.7
Negative	953	81.3

^§^ Clinical relevance (positive) if score “moderate–severe” ≥14; ^§§^ Clinical relevance (positive) if score “moderate–severe” ≥10; ^§§§^ Clinical relevance (positive) if score “moderate–severe” ≥19; ^§§§§^ Clinical relevance (positive) if score “moderate–severe” ≥ 33.

**Table 2 ijerph-19-07313-t002:** Statistically significant associations with the psychometric results (univariate regressions).

**(a)** Positive DASS-21 Depression	OR	95% CI	*p*-Value
Gender			0.002
Female	REF	REF	
Male	0.59	0.42; 0.83	
Professional role			0.038
Other managers	REF	REF	
Nurses, sanitary technicians	2.31	1.03; 5.20	
Administration	1.68	0.68; 4.19	
Physicians and surgeons	2.60	1.15; 5.86	
Changes in use of alcoholics			0.001
No change	REF	REF	
Decreased	1.00	0.54; 1.85	
Increased	2.32	1.53; 3.51	
Changes in smoking			0.000
No change	REF	REF	
Decreased	1.07	0.28; 4.10	
Increased	2.86	1.57; 5.20	
Changes in eating habits			0.000
No change	REF	REF	
More regular	1.05	0.67; 1.65	
More disordered	3.81	2.78; 5.21	
**(b)** **Positive DASS-21 anxiety**			
Professional role			0.000
Other managers	REF	REF	
Nurses, sanitary technicians	4.04	1.71; 9.56	
Administration	2.97	1.17; 7.59	
Physicians and surgeons	2.21	0.92; 5.29	
Being in charge of COVID-19 patients			0.021
No	REF	REF	
Yes (“front-line HCWs”)	1.54	1.08; 2.21	
Educational level			0.004
Less than high-school	REF	REF	
High-school	1.13	0.54; 2.38	
University Diploma/ Bachelor’s Degree	1.12	0.49; 2.55	
University Degree/Master’s Degree/Specialisation	0.66	0.32; 0.73	
Increase in workload			0.003
No	REF	REF	-
Yes	1.52	1.15; 2.02	-
Changes in use of alcoholics	-	-	0.001
No change	REF	REF	-
Decreased	0.39	0.18; 1.85	-
Increased	1.90	1.26; 2.88	-
Changes in smoking	-	-	0.001
No change	REF	REF	-
Decreased	1.23	0.36; 4.15	-
Increased	2.25	1.27; 4.01	-
Changes in eating habits	-	-	0.000
No change	REF	REF	-
More regular	1.29	0.86; 1.94	-
More disordered	3.09	2.27; 4.21	-
**(c)** **Positive DASS-21 stress**			
Gender			0.028
Female	REF	REF	
Male	0.72	0.54; 0.97	
Professional role			0.000
Other managers	REF	REF	
Nurses, sanitary technicians	2.77	1.29; 5.94	
Administration	1.53	0.64; 3.65	
Physicians and surgeons	3.32	1.55; 7.13	
Organizational Area			0.017
Surgical Area	REF	REF	
Direction and Staff	0.60	0.38; 0.95	
Emergency Area	1.26	0.79; 2.02	
Medical Area	0.96	0.63; 1.44	
Services Area	0.98	0.58; 1.63	
Being in charge of COVID-19 patients			0.021
No	REF	REF	
Yes (“front-line HCWs”)	1.51	1.0; 2.13	
Increase in workload			0.000
No	REF	REF	
Yes	1.71	1.31; 2.24	
Having direct contact with COVID-19 patients			0.014
No	REF	REF	
Yes	1.38	1.06; 1.79	
Changes in use of alcoholics			0.000
No change	REF	REF	
Decreased	0.49	0.25; 0.94	
Increased	2.25	1.52; 3.33	
Changes in smoking			0.001
No change	REF	REF	
Decreased	0.51	0.11; 2.44	
Increased	2.76	1.56; 4.89	
Changes in eating habits			0.000
No change	REF	REF	
More regular	1.16	0.78; 1.72	
More disordered	3.94	2.93; 5.29	
**(d)** **Positive IES-R**			
Gender			0.001
Female	REF	REF	
Male	0.54	0.38; 0.77	
Professional role			0.003
Other managers	REF	REF	
Nurses, sanitary technicians	3.62	1.42; 9.22	
Administration	2.71	0.98; 7.05	
Physicians and surgeons	2.39	0.93; 6.14	
Educational level			0.0014
Less than high-school	REF	REF	
High-school	1.50	0.60; 3.78	
University Diploma/Bachelor’s Degree	2.52	0.96; 6.66	
University Degree/Master’s Degree/Specialisation	1.19	0.49; 2.89	
Increase in workload			0.004
No	REF	REF	
Yes	1.55	1.14; 2.10	
Changes in use of alcoholics			0.001
No change	REF	REF	
Decreased	0.36	0.14; 0.91	
Increased	1.91	1.24; 2.97	
Changes in smoking			0.004
No change	REF	REF	
Decreased	0.69	0.14; 3.29	
Increased	2.23	1.20; 4.12	
Changes in eating habits			0.000
No change	REF	REF	
More regular	1.05	0.64; 1.71	
More disordered	4.42	3.18; 6.14	

## Data Availability

The data presented in this study are available on request from the corresponding author; they are not publicly available due to restrictions concerning privacy and ethical reasons.

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
