# Peer review of "“It’s All COVID’s Fault!”: Symptoms of Distress among Workers in an Italian General Hospital during the Pandemic"

_ijerph, 2022, doi:10.3390/ijerph19127313_

Round 1
Reviewer 1 Report
Dear Authors,
Please find below my feedback to your manuscript "It’s all COVID’s fault!”: symptoms of distress among workers in an Italian general hospital during the pandemic".
Abstract
- You start the abstract referring to the pandemic in the present tense, then move onto discussing the aims, methods and conclusions of the study in past tense. It would make much more sense grammatically to use the past tense throughout.
Introduction
- Lines 131-137 appear disjointed from the rest of the introduction. It would be good to signpost with a simple introductory phrase such as: "Through our review of the current evidence, we identified some gaps in the knowledge that enabled us to define the purposes of our study:
- Primary objective.... "
Materials and methods
- The scales chosen and the way they were used are appropriate.
Results
- Excellent questions in your survey, tapping into important issues such as change of eating habits that are themselves associated with mental health outcomes.
- The results are presented clearly. However, the number of tables you have used may be overwhelming for some readers. If you considered presenting some of the results in a simple diagrammatic / visual manner, e.g. a simple bar or pie chart, this would add substantial value to your manuscript.
Discussion
- Your discussion of the results is excellent.
Conclusion
- Your conclusion lacks a clear link to your study population. It would benefit from one sentence that makes it clear to the reader who your participants were and why this is relevant.
Congratulations on a really interesting study. I wish you all the very best with the rest of the review process.
Kindest regards,
The reviewer
Reviewer 2 Report
Results
|
It presents multiple results in its tables that could be simplified by collecting the most important information.
Some results are confusing, e.g: Table 3. Statistically significant associations to psychometric results (univariate regressions)
[…] Professional role, p-value: 0.038, Nurses, sanitary technicians, Administration, Physicians and surgeons[…]
No differences between different professional positions? e.g:
Table 4. Associations between gender and working with COVID+ patients […]
There is no analyze between different professionals positions. Working in COVID+ y COVID- Only analyzed by gender
|
243
256 |
Author Response
Reply to Referee 2
- It presents multiple results in its tables that could be simplified by collecting the most important information.
- Some results are confusing, e.g:
Table 3. Statistically significant associations to psychometric results (univariate regressions)
[…] Professional role, p-value: 0.038, Nurses, sanitary technicians, Administration, Physicians and surgeons[…]
No differences between different professional positions? e.g:
Table 4. Associations between gender and working with COVID+ patients […]
There is no analyze between different professionals positions. Working in COVID+ y COVID- Only analyzed by gender
We agree with the reviewer that the amount of data displayed in the tables was excessive and confusing. Therefore, we have decided to reduce the tables to only two of the original four, by eliminating not necessary information and by compacting together table 1 and 2, and table 3 and 4. Also, two graphic representations of data were included (figures # 1 and 2) to guide and support reading.
Please see attachment

Reviewer 3 Report
The manuscript describes statistics from a survey administered to the whole staff of the Modena General University Hospital three months after the onset of the pandemic, in 2020. Demographics and changes in working and living conditions were used to predict binary status from Depression, Anxiety and Stress Scale (DASS-21) and the Impact of Event Scale-Revised (IES-R). Factors including gender, job role, ward, changes in lifestyle, first-line work, significantly predict group assignment. The data might be valuable but the statistical approach is confusing. I'm not sure why the continuous depression scales had to be dichotomized. Wouldn't an ordinary linear regression do a better job by keeping the fine-grained information in DASS-21 and IES-R? Moreover, if we were to use logistic regression to predict binary group assignment, I would expect to include all variables as the multi-variate input instead of a univariate regression, otherwise, there is no point in using logistic regression (a simple univariate t-test would do the job). I'm also confused about how multiple comparisons were corrected and whether covariates were considered in each logistic regression model. Lastly, I'm confused about why Table 4 is separated from Table 3. Are they using different statistical models?
Author Response
Reply to Referee 3
The manuscript describes statistics from a survey administered to the whole staff of the Modena General University Hospital three months after the onset of the pandemic, in 2020. Demographics and changes in working and living conditions were used to predict binary status from Depression, Anxiety and Stress Scale (DASS-21) and the Impact of Event Scale-Revised (IES-R). Factors including gender, job role, ward, changes in lifestyle, first-line work, significantly predict group assignment.
- The data might be valuable but the statistical approach is confusing. I'm not sure why the continuous depression scales had to be dichotomized. Wouldn't an ordinary linear regression do a better job by keeping the fine-grained information in DASS-21 and IES-R?
- Moreover, if we were to use logistic regression to predict binary group assignment, I would expect to include all variables as the multi-variate input instead of a univariate regression, otherwise, there is no point in using logistic regression (a simple univariate t-test would do the job).
The reviewer is right, DASS-21 and IES-R are commonly used as continuous variables and the calculation of average scores and standard deviations at the three subscales has been included (lines 220-221). Nevertheless, their clinical dichotomized versions are also commonly used (e.g. https://www.frontiersin.org/articles/10.3389/fpsyt.2021.574671/full; https://doi.org/10.1007/s00737-021-01145-0; https://doi.org/10.1186/s12884-020-03399-5), to emphasize the impact in terms of clinical differences within the study population. Basically, we wanted to focus on what could make a difference between those subjects more suffering and the others.
Moreover, the choice to use dichotomization of data, rather than maintaining the 4 original classes, contributed to a good statistical performance of the logistic regressions in terms of significance and accuracy.
As further analyses of the data are planned, we will certainly also perform these by using DASS and IES as continuous variables.
Finally, as we chose, for now, to only test categorized variables, a simple univariate t-test was not applicable.
- I'm also confused about how multiple comparisons were corrected and whether covariates were considered in each logistic regression model.
The reviewer is very right in noticing this, and we would like to explain that no covariates were considered in the logistic regression models; also, the results here displayed were not corrected. We have included in the limitation section a more detailed explanation of this, together with a clearer declaration of the exploratory nature of the present study.
- Lastly, I'm confused about why Table 4 is separated from Table 3. Are they using different statistical models?
Table 4 was included to support the idea to focus on gender and working at the front-line as relevant variables impacting on emotional distress (as it is the case), but the reviewer is right in pointing out that this distinction results in confusion and redundancy. Table 4 was removed and information better detailed in table 3.
Please see attachment
